# Prediction of Solid Soluble Content of Green Plum Based on Improved CatBoost

Xiao Zhang [1,2], Chenxin Zhou [1], Qi Sun [1], Ying Liu [1,*], Yutu Yang [1] and Zilong Zhuang [1]

[1] College of Mechanical and Electronic Engineering, Nanjing Forestry University, Nanjing 210037, China; zx_xhx@njfu.edu.cn (X.Z.); zhouchx@njfu.edu.cn (C.Z.); supremesunqi@njfu.edu.cn (Q.S.); yangyutu@njfu.edu.cn (Y.Y.); zzl0702@njfu.edu.cn (Z.Z.)

[2] Nanjing Institute of Agricultural Mechanization, Ministry of Agriculture and Rural Affairs, Nanjing 210014, China

[*] Correspondence: liuying@njfu.edu.cn

**Abstract:** Most green plums need to be processed before consumption, and due to personal subjective factors, manual harvesting and sorting are difficult to achieve using standardized processing. Soluble solid content (SSC) of green plum was taken as the research object in this paper. Visible near-infrared (VIS-NIR) and shortwave near-infrared (SW-NIR) full-spectrum spectral information of green plums were collected, and the spectral data were corrected and pre-processed. Random forest algorithm based on induced random selection (IRS-RF) was proposed to screen four sets of characteristic wavebands. Bayesian optimization CatBoost model (BO-CatBoost) was constructed to predict SSC value of green plums. The experimental results showed that the preprocessing method of multiplicative scatter corrections (MSC) was obviously superior to Savitzky–Golay (S–G), the prediction effect of SSC based on VIS-NIR spectral waveband by partial least squares regression model (PLSR) was obviously superior to SW-NIR spectral waveband, MSC + IRS-RF was obviously superior to corresponding combination of correlation coefficient method (CCM), successive projections algorithm (SPA), competitive adaptive reweighted sampling (CARS), and random forest (RF). With the lowest dimensional selected feature waveband, the lowest VIS-NIR band group was only 53, and the SW-NIR band group was only 100. The model proposed in this paper based on MSC + IRS-RF + BO-CatBoost was superior to PLSR, XGBoost, and CatBoost in predicting SSC, with $R^2_P$ of 0.957, which was 3.1% higher than the traditional PLSR.

**Keywords:** green plum; spectral technique; SSC; BO-CatBoost; feature band groups

## 1. Introduction

Most green plums need to be processed before consumption. Due to different components of each green plum, the processed products are also different [1,2]. Green plums with high acidity and low sugar content are usually used to make green plum essence, while those with high sugar content and low acidity are usually used to make green plum wine, etc. The component content of green plums will vary with different maturity levels [3].

When manually picking and sorting green plums, the main basis is the skin color and picking time of green plums, and the composition content is determined and classified based on manual experience. However, due to personal subjective factors, it is difficult to achieve standardized processing. Traditional methods for determining the acidity and sugar content of green plums are destructive and inefficient [4,5]. Therefore, research on new non-destructive methods for detecting the composition of green plums is of great significance for improving the processing efficiency of green plum.

SSC is one of the important reference indicators for measuring the maturity, internal quality, and edible processing characteristics of fruits. The experimental results showed that as the SSC index increased, the maturity of fruits increased. Therefore, many experts

and scholars conducted non-destructive testing research on the maturity of fruits such as apples, pears, grapes, strawberries, and watermelons based on SSC prediction.

Currently, spectroscopic techniques based on spectral features such as near-infrared spectroscopy and hyperspectral imaging became the main means of non-destructive detection technology [6–10]. Ma T et al. used VIS-NIR spectroscopy to predict SSC in apples with a determination coefficient $R^2$ and root mean square error (RMSE) of 0.97 and 0.20% [11]. Yu X et al. combined the hyperspectral imaging with a deep learning method consisting of stacked autoencoders (SAE) and fully connected neural networks (FNN) to predict SSC in postharvest Kurele pears, with a determination coefficient $R^2$ and RMSE of 0.92 and 0.22% [12]. The hyperspectral technology was used to predict SSC in netted melons by continuous wavelet transformation. The correlation coefficient and RMSE of the random forest regression model decomposed by the continuous wavelet transform were 0.72 and 0.98%, respectively [13].

The aforementioned research methods relied on collecting full-spectrum spectral information to predict the internal different component contents of the fruit and achieved a high prediction accuracy. However, if high-spectrum equipment is used to build green plums sorting production line, there are still problems such as long-time consumption, high cost, and difficulty in practical promotion and application [14,15]. When using traditional multispectral technology to select specific waveband groups and predict the internal different component contents by building PLSR, support vector regression (SVR), and other traditional machine learning models, the prediction accuracy cannot meet the actual sorting requirements. Liu C et al. established a model to predict dicyandiamide (DCD) in milk powder based on multispectral technology using partial least squares (PLS), least squares support vector machine (LS-SVM), and backpropagation neural network (BPNN) with $R^2$ of 0.873 [16]. Younas et al. combined multispectral imaging technology with chemical metrology to build a relationship between multispectral images and water content in mushrooms using PLSR, BPNN, and LS-SVM models, with the highest $R^2$ figure reaching 0.86 [17]. Chakravartula et al. used the 1940 nm, 1500 nm, and 2050 nm wavebands to build principal components analysis-partial least squares regression model (PCA-PLSR) to detect water content, protein, and other indicators in bread, with $R^2$ reaching 0.88 at the highest. Currently, research on predicting different component contents of green plums using multispectral technology is still blank [18].

To resolve the problems of time-consuming, high cost, and difficulty in practical application of non-destructive testing technology for green plums, SSC prediction of green plums was taken as the research object in this study. VIS-NIR and SW-NIR full-spectrum spectral information were collected from green plum samples, and the spectral data were corrected and pre-processed. An IRS-RF algorithm was proposed to screen the characteristic bands, and a BO-CatBoost model was constructed to predict SSC value of different green plum samples. This makes model prediction accuracy meet the actual sorting requirements. The selected feature band groups provide a theoretical basis for future multi-spectral technology based on green plum sorting research. The technical route is shown in Figure 1.

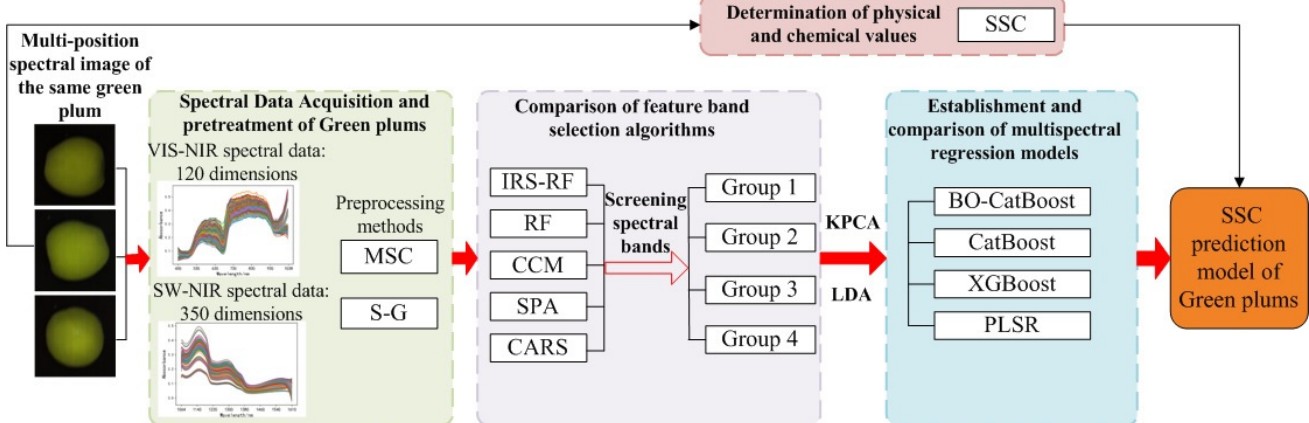

**Figure 1.** Technical route of subject research.

## 2. Materials and Methods

### 2.1. Spectrum Data Collection

2.1.1. Sample Sources

In May 2021, a total of 276 Zhao Shui green plum samples were purchased and screened from Yunnan Province, China, for the purpose of predicting SSC value of green plums. The samples were divided into a training set and a test set in a ratio of about 4:1, with 221 samples as the training set and 55 samples as the test set.

The samples were placed in a laboratory refrigerator at a constant temperature of 4 °C. Samples were randomly selected for each experiment, placed in advance at ambient temperature, and spectral data were collected and SSC value determination of each green plum was performed when their temperature was the same as room temperature.

2.1.2. Hardware Composition of Hyperspectral Acquisition Device

A non-destructive hyperspectral imaging system for predicting SSC of green plum samples was set up (Figure 2), comprised of a push-broom VIS-NIR hyperspectral camera (GaiaField-V10E-AZ4, Jiangsu Dualix Spectral Image Technology Co., Ltd., Wuxi, China), a push-broom SW-NIR hyperspectral camera (GaiaField-N17E-HR, Jiangsu Dualix Spectral Image Technology Co., Ltd., China), two self-made dome light source systems, an uninterrupted power supply (UPS) (C3K, Shante, Hangzhou, China), a transmission desk, a dark chamber, and a computer (T570, Lenovo, Beijing, China). Each camera was equipped with the same dome light source system, which included 12 halogen lamps (Halogen 12V, Philips, Suzhou, China), with UPS providing a stable power supply. The whole system was surrounded by the dark chamber to prevent external light interference.

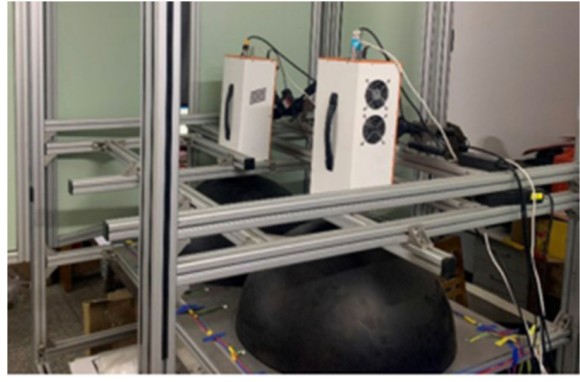

**Figure 2.** Actual photo of hyperspectral imaging acquisition system.

During the high-spectrum image acquisition stage, the green plum samples were derived evenly to below the high-spectrum camera by the conveyor belt, and the conveyor belt speed was set to match the acquisition speed of the high-spectrum camera. The high-spectrum acquisition system is shown in Figure 2.

Before the experiment, the hyperspectral imaging system should be turned on and warmed up for 30 min. The optimal parameters for spectral acquisition was determined by pre-test: exposure time and conveyor belt moving speed. After the green plum sample was scanned, the obtained hyperspectral data were calibrated by black and white board. The calibrated image ($A_0$) was calculated using the following Formula (1):

$$A_0 = \frac{A - A_D}{A_W - A_D} \times 100\% \tag{1}$$

wherein $A_0$ represents the green plums spectrum reflectance data after black and white calibration, $A$ is the original green plums spectrum data to be corrected, $A_D$ represents the spectral reflectance data collected with the lens cover on, and $A_W$ represents the spectral data of the standard 99% reflectance plate.

After spectral data were collected, the green plums were squeezed to extract green plum juice immediately. The PAL-1 hand-held refractometer was used to measure the SSC of green plums. The measurement range was 0.0–53.0% BRIX, with an accuracy of $\pm0.2\%$ BRIX. The sample tank should be cleaned before measuring, the green plum sample was then squeezed into juice. After precipitating, the supernatant was dropped into the sample tank and the SSC value was recorded.

The SSC values of 276 green plum samples are shown in Table 1.

**Table 1.** SSC values of green plum samples.

| Sample Set | Number of Samples | Minimum Value | Maximum Value | Average Value |
|---|---|---|---|---|
| Training set | 221 | 5.8 | 13.1 | 9.7263 |
| Test set | 55 | 5.8 | 12.4 | 9.7033 |

### 2.2. Preprocessing of Spectral Data

All the tested software and hardware configurations as well as compilation environments are shown in Table 2.

**Table 2.** Software and hardware environment configuration.

| Name | Parameters |
|---|---|
| System | Windows 10 × 64 |
| CPU | Inter I9 9900K@3.60 GHz |
| GPU | Nvidia GeForce RTX 2080 Ti (11 G) |
| Environment configuration | PyCharm + Pytorch 1.7.1 + Python 3.7.7 Cuda 10.2 + cudnn 7.6.5 + tensorboardX 2.1 |

The hyperspectral imaging system was used to collect the images of green plum samples, the VIS-NIR spectrum range was 400–1000 nm, with 120 data channels, and the SW-NIR spectrum range was 900–1700 nm, with 350 data channels. In order to reduce the error caused by the spectral reflection due to a single posture, the spectral images of three different postures of each green plum sample was extracted and the average was taken from three positions, as shown in Figure 3.

ENVI5.3 was used to determine the region of interest (ROI) of the images, and extract the average spectrum of the green plum samples in the ROI as the original spectrum data, and Figure 4 is the original spectrum reflectance curve of all green plum samples. Different colored lines represent the spectral characteristic curves of different green plum samples, with a total of 276 lines.

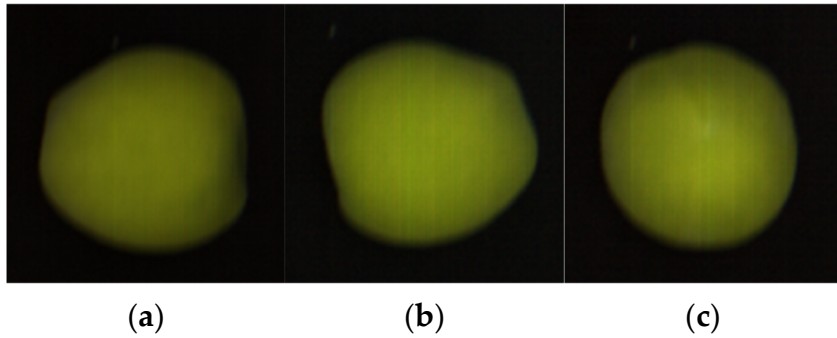

**Figure 3.** Spectral images of three different positions of the same green plum: (**a**) left posture; (**b**) right posture; (**c**) top posture.

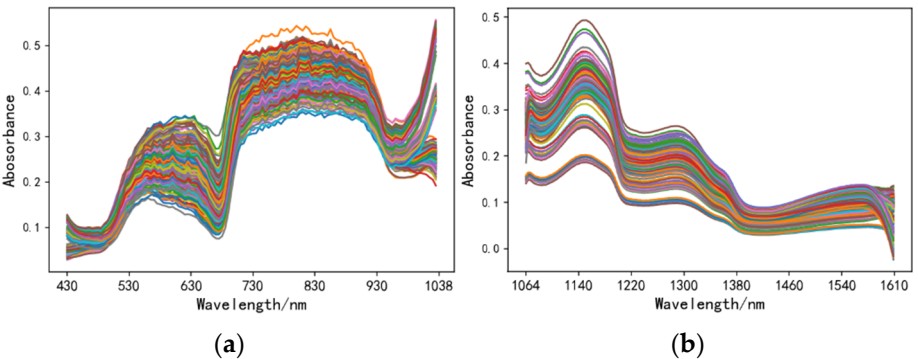

(**a**)  (**b**)

**Figure 4.** Original spectral reflectance curve of green plum samples: (**a**) VIS-NIR (**b**) SW-NIR.

*2.3. Improved Random Forest Feature Extraction Based on Induced Random Selection*

An IRS-RF algorithm was proposed in this paper, which measured the importance of all features, clustered them according to their weight values using the K-means clustering algorithm and, finally, selected the feature subspace from each class to construct the decision tree based on the partition ratio. This algorithm could reduce the probability of highly correlated features being selected at the same time, decrease the uncertainty of node splitting and the correlation between decision trees, increase the diversity of decision trees, and achieve a smaller generalization error. The steps were as follows:

(1)  Measure the importance of features

Calculate the correlation coefficient *r* between spectral features according to Equation (2), and use it as the feature importance weight.

$$r = \frac{\sum(X - \overline{X})(Y - \overline{Y})}{\sqrt{\sum(X - \overline{X})^2 \times \sum(Y - \overline{Y})^2}} \qquad (2)$$

(2)  K-means clustering for feature classification

Randomly select *k* data points as the initial clustering centers for *k* clusters, and each data point is divided into the closest cluster to it to form the initial distribution of *k* clusters. For each allocated cluster, recalculate their respective cluster centers and iterate multiple times until the cluster centers remain unchanged. Using correlation as the importance weight of a feature, it is divided into *k* feature regions with varying degrees of importance.

(3)  Proportional sampling

Feature selection is selected and constructed to build a decision tree in a certain proportion. The number of randomly selected features in the *i*-th feature area is calculated according to Equation (3), and $N_1$, $N_2$, ... $N_k$ features are randomly selected from *k* feature areas to form the feature subspace of this tree. Select features proportionally in different

feature intervals, i.e., perform induced random selection, which makes the feature subspace more representative.

$$N_i = m_{try}\frac{m_i}{m} \tag{3}$$

Among them, $N_i$ represents the number of features extracted from the *i*-th feature area, *m* represents the total number of features, $m_i$ represents the number of features in the *i*-th feature area, and the number of feature variables in each feature tree.

## 3. BO-CatBoost Model Based on Bayesian Optimization Algorithm

With the advancement in computing power, models are becoming increasingly complex. To ensure that the model does not fall into a local minimum and to avoid excessive computation, a Bayesian optimization algorithm can be used. The core components of Bayesian optimization are a statistical description proxy model and an acquisition function [19–21]. In the proxy function model, a flexible surrogate model was used to randomly approximate the target function, which was difficult to calculate, and different kernel functions were used to increase the nonlinear expression ability of the proxy model. The acquisition function balances the development of high mean regions and the exploration of high volatility regions to select suitable hyperparameter sample points.

The Bayesian optimization classification algorithm, using the classical Gaussian process as a proxy model, was introduced in this paper, and an improved BO-CatBoost algorithm was built to ensure the stability and friendliness of the model while gradually improving the performance of the algorithm. BO-CatBoost algorithm flowchart is shown in Figure 5. The main steps were as follows:

(1) Initialize the Bayesian optimization algorithm point set and the maximum number of iterations $N$;

(2) Based on the current set of points, build the Gaussian process proxy function;

(3) Based on the proxy function, maximize the acquisition function to obtain the next evaluation point;

(4) Obtain the evaluation point $x_t$ function value $f(x_t)$, add it to the evaluation point set;

(5) Termination condition determination: if the number of iterations meets the default criteria, stop searching or return to step 2 for further search;

(6) After iteration, obtain the optimal BO-CatBoost parameters, and use the optimal parameters to study and model the training data;

(7) Finally, test the model with the test set, output the evaluation result.

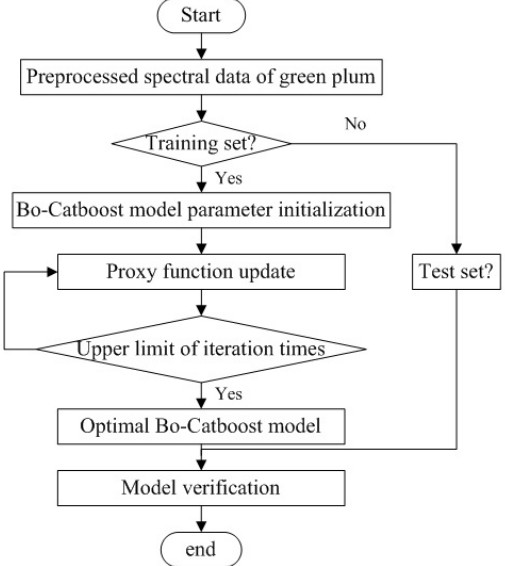

**Figure 5.** BO-CatBoost algorithm flowchart.

## 4. Results and Discussion

### 4.1. Model Training and Result Analysis

The prediction coefficient $R^2$, mean absolute error (MAE), and RMSE were selected as the model performance evaluation indicators. The smaller MAE and RMSE values and the larger the $R^2$ value, the better model performance and prediction effect. The evaluation indicators of the training set were represented by $R^2_C$, MAEC, and RMSEC, respectively, and the test set were $R^2_P$, MAEP, and RMSEP. The PLSR was built and used to predict SSC of green plums under different pre-processing and feature extraction methods were compared.

MSC and S–G were used to pre-process the spectral data of VIS-NIR and SW-NIR, to remove noise and invalid information. The pre-processed SW-NIR and VIS-NIR spectral band data are shown in Figure 6. Different colored lines represent the spectral characteristic curves of different green plum samples, with a total of 276 lines.

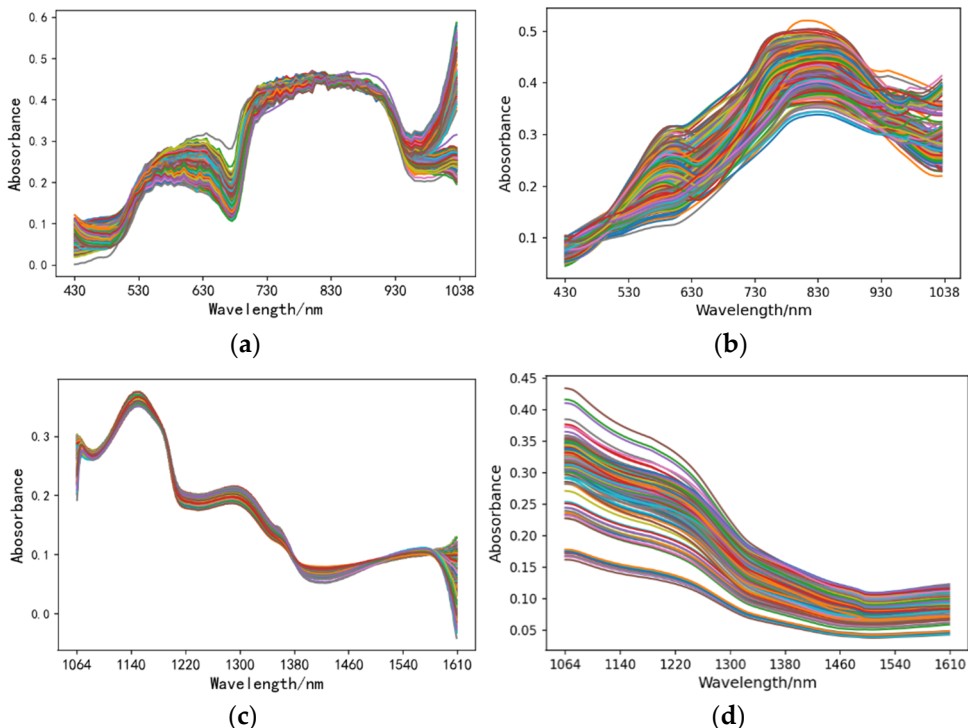

**Figure 6.** Comparison of the effects of different preprocessing methods for spectral data. (**a**) VIS-NIR spectral data preprocessed by MSC. (**b**) VIS-NIR spectral data preprocessed by S–G. (**c**) SW-NIR spectral data preprocessed by MSC. (**d**) SW-NIR spectral data preprocessed by S–G.

The absorption of the spectrum mainly reflects information of hydrogen groups such as C-H, O-H, and N-H in organic substances, while SSC contains important information of the O-H group. As shown in Figure 6, MSC could effectively eliminate spectral differences caused by different scattering levels, while the effect of difference was not good after S–G pre-processing. After MSC pre-processing, a noticeable absorption peak produced by the O-H bond stretching vibration was present at 730 nm in the VIS-NIR spectrum of green plums, the subsequent decrease to 930 nm may have been influenced by the quadruple frequency stretching vibration of C-H. In the SW-NIR spectrum, the decline from 1140–1220 nm may be caused by the first frequency absorption of the N-H group. The differences in the VIS-NIR and SW-NIR spectra were significant, which may have been caused by stronger interference from moisture in the SW-NIR range. In summary, MSC pre-processing could enhance the correlation between spectrum and data.

### 4.1.1. Comparison of SSC Prediction Results with Different Pre-Processing Methods and Feature Extraction Combination Algorithms

Based on different pre-processing methods and feature extraction algorithm combinations, the prediction performance results of PLSR models based on VIS-NIR and SW-NIR spectra are shown in Table 3.

**Table 3.** Influence of different preprocessing and feature extraction algorithms on SSC prediction.

| Bands | Preprocessing + Characteristic Wavelength Combination Algorithm | RMSEC | MAEC | $R^2_C$ | RMSEP | MAEP | $R^2_P$ |
|---|---|---|---|---|---|---|---|
| VIS-NIR Spectral Band | MSC + CCM | 0.401 | 0.323 | 0.909 | 0.420 | 0.339 | 0.901 |
| | MSC + SPA | 0.369 | 0.305 | 0.925 | 0.378 | 0.316 | 0.920 |
| | MSC + CARS | 0.605 | 0.479 | 0.827 | 0.614 | 0.490 | 0.814 |
| | MSC + RF | 0.418 | 0.321 | 0.918 | 0.428 | 0.327 | 0.914 |
| | MSC + our algorithm | 0.341 | 0.255 | 0.933 | 0.359 | 0.261 | 0.928 |
| | S–G + CCM | 0.376 | 0.276 | 0.912 | 0.380 | 0.288 | 0.902 |
| | S–G + SPA | 0.391 | 0.303 | 0.894 | 0.405 | 0.319 | 0.889 |
| | S–G + CARS | 0.614 | 0.465 | 0.738 | 0.639 | 0.475 | 0.723 |
| | S–G + RF | 0.607 | 0.447 | 0.759 | 0.611 | 0.458 | 0.747 |
| | S–G + our algorithm | 0.456 | 0.258 | 0.916 | 0.467 | 0.268 | 0.905 |
| SW-NIR Spectral Band | MSC + CCM | 0.562 | 0.451 | 0.839 | 0.572 | 0.455 | 0.827 |
| | MSC + SPA | 0.531 | 0.408 | 0.871 | 0.543 | 0.423 | 0.851 |
| | MSC + CARS | 0.752 | 0.599 | 0.798 | 0.786 | 0.615 | 0.785 |
| | MSC + RF | 0.661 | 0.402 | 0.822 | 0.670 | 0.410 | 0.805 |
| | MSC + our algorithm | 0.473 | 0.257 | 0.925 | 0.488 | 0.266 | 0.911 |
| | S–G + CCM | 0.637 | 0.461 | 0.732 | 0.649 | 0.480 | 0.715 |
| | S–G + SPA | 0.457 | 0.321 | 0.873 | 0.462 | 0.334 | 0.855 |
| | S–G + CARS | 0.552 | 0.359 | 0.790 | 0.569 | 0.371 | 0.781 |
| | S–G + RF | 0.718 | 0.525 | 0.652 | 0.738 | 0.544 | 0.631 |
| | S–G + our algorithm | 0.490 | 0.301 | 0.903 | 0.495 | 0.304 | 0.892 |

As can be seen from the data in Table 3, for the VIS-NIR range, the $R^2_P$ of MSC combined with five different feature extraction algorithms were 0.901, 0.920, 0.814, 0.914, and 0.928, respectively, all of which were higher than the corresponding $R^2_P$ of S–G, which were 0.902, 0.889, 0.723, 0.747, and 0.905; for the SW-NIR range, the $R^2_P$ of MSC combined with five different feature extraction algorithms were 0.827, 0.851, 0.785, 0.805, and 0.911, respectively, all of which were higher than the corresponding $R^2_P$ of S–G, which were 0.715, 0.855, 0.781, 0.631, and 0.892. It can be seen that the $R^2_P$ corresponding to the MSC pre-processing method was obviously better than the $R^2_P$ corresponding to the S–G. In conclusion, MSC was selected as the pre-processing method for predicting SSC of green plums.

The improved IRS-RF algorithm was used to extracts four sets of feature band groups in this paper and compared with CCM, SPA, CARS, and RF algorithms. The effects of different feature extraction algorithms on the SSC test set are shown in Table 4.

As can be seen from the data in Table 4, in terms of prediction accuracy: for the VIS-NIR spectral band, the RMSEP and MAEP of MSC + IRS-RF (our algorithm) were the lowest, only 0.359 and 0.261, respectively, with the highest $R^2_P$ of 0.928. For the SW-NIR spectral band, the RMSEP and MAEP of MSC + IRS-RF were the lowest, only 0.488 and 0.266, respectively, with the highest $R^2_P$ of 0.911. The prediction performance of SSC in the VIS-NIR spectral band was obviously better than that in the SW-NIR spectral band. In terms of spectral dimension, for the VIS-NIR spectral band, the number of selected spectral bands by IRS-RF was only 53. For the SW-NIR spectral band, the number of selected spectral bands by IRS-RF was only 100. The IRS-RF algorithm measured the importance of all features, reducing the probability of features with high correlation being selected simultaneously. Therefore, the number of selected spectral bands by MSC + IRS-RF was the smallest.

**Table 4.** Influence of different feature extraction algorithms on SSC prediction.

| Algorithms | Bands | Characteristic Band Group Wavelength (nm) | | | | Dimension | RMSEP | MAEP | $R^{2.5}{}_P$ |
|---|---|---|---|---|---|---|---|---|---|
| | | Group 1 | Group 2 | Group 3 | Group 4 | | | | |
| CCM | | 430–488 | 636–666 | 702–925 | 968–1038 | 79 | 0.420 | 0.339 | 0.901 |
| SPA | VIS-NIR | 440–524 | 596–646 | 707–846 | 957–1038 | 74 | 0.378 | 0.316 | 0.920 |
| CARS | Spectral | 450–542 | 596–646 | 707–867 | 957–1038 | 80 | 0.614 | 0.490 | 0.814 |
| RF | Band | 430–483 | 631–656 | 707–925 | 973–1038 | 75 | 0.428 | 0.327 | 0.914 |
| Ours | | 430–498 | 552–608 | 756–798 | 952–1038 | 53 | 0.359 | 0.261 | 0.928 |
| CCM | | 1167–1239 | 1253–1358 | 1489–1573 | 1601–1610 | 159 | 0.572 | 0.455 | 0.827 |
| SPA | SW-NIR | 1066–1150 | 1165–1333 | 1367–1467 | 1518–1568 | 236 | 0.543 | 0.423 | 0.851 |
| CARS | Spectral | 1068–1149 | 1159–1353 | 1363–1476 | 1576–1610 | 248 | 0.786 | 0.615 | 0.785 |
| RF | Band | 1024–1150 | 1164–1350 | 1507–1568 | 1591–1610 | 231 | 0.670 | 0.410 | 0.805 |
| Ours | | 1182–1226 | 1281–1345 | 1526–1573 | 1595–1610 | 100 | 0.488 | 0.266 | 0.911 |

In conclusion, preliminary selection of MSC + IRS-RF for pre-processing and spectral band selection of VIS-NIR spectral band was selected. Figure 7 compares the prediction results of SSC values in different spectral bands of VIS-NIR and SW-NIR based on the MSC + IRS-RF algorithm.

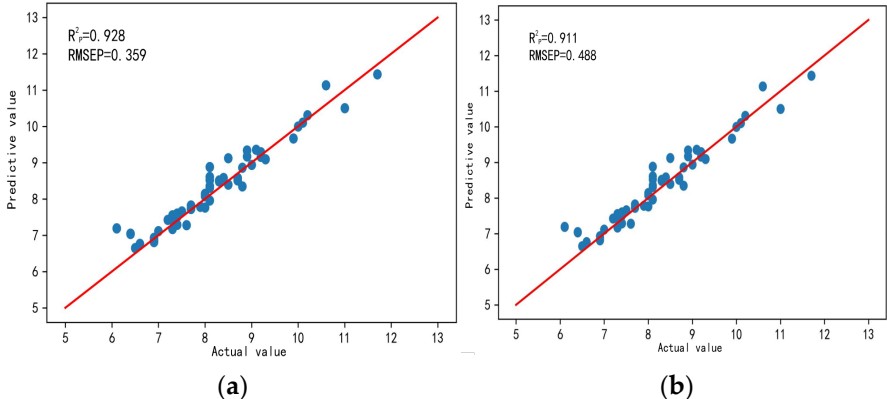

(**a**)　　　　　　　　　　　　　　　　　　　　(**b**)

**Figure 7.** Prediction of SSC results of different spectral bands based on MSC + IRS-RF: (**a**) VIS-NIR spectral band; (**b**) SW-NIR spectral band.

4.1.2. Comparison of SSC Prediction Results of Different Machine Learning Regression Models

Based on the research foundation in 4.1.1, the MSC + IRS-RF model was used for preprocessing and feature wavelength selection of the VIS-NIR spectral bands. In this paper, we proposed a BO-CatBoost (our algorithm) model based on Bayesian optimization algorithm with a learning rate of 0.1 and 500 iterations. Table 5 compares the prediction results of our algorithm with different regression models including conventional PLSR, XGBoost, and CatBoost for the SSC of green plums.

**Table 5.** Influence of different regression models on SSC prediction.

| Regression Model | Characteristic Band Group Wavelength (nm) | | | | Dimension | RMSEP | MAEP | $R^2{}_P$ |
|---|---|---|---|---|---|---|---|---|
| | Group 1 | Group 2 | Group 3 | Group 4 | | | | |
| PLSR | 430–498 | 552–608 | 756–798 | 952–1038 | 53 | 0.359 | 0.261 | 0.928 |
| XGBoost | 432–498 | 556–592 | 756–798 | 966–1028 | 43 | 0.403 | 0.191 | 0.927 |
| CatBoost | 432–498 | 562–592 | 742–786 | 972–1018 | 39 | 0.365 | 0.231 | 0.942 |
| Ours | 452–498 | 538–566 | 756–782 | 982–1032 | 31 | 0.252 | 0.189 | 0.957 |

The experimental results showed that the $R^2_P$ of PLSR, XGBoost, and CatBoost regression models were 0.928, 0.927, and 0.942, respectively. The $R^2_P$ of BO-CatBoost (our algorithm) was the highest, reaching 0.957 with the lowest RMSEP and MAEP of 0.252 and 0.189, respectively. It was improved by 3.1% compared to traditional PLSR model. The selected four feature wavelength dimensions were the lowest with only 31, less than 53, 43, and 39 of PLSR, XGBoost, and CatBoost, respectively. The wavelength ranges were 452–498 nm, 538–566 nm, 756–782 nm, and 982–1032 nm. Figures 8 and 9 compare the prediction results of different regression models for the SSC of green plums and the selected feature wavelength groups for SSC prediction, respectively.

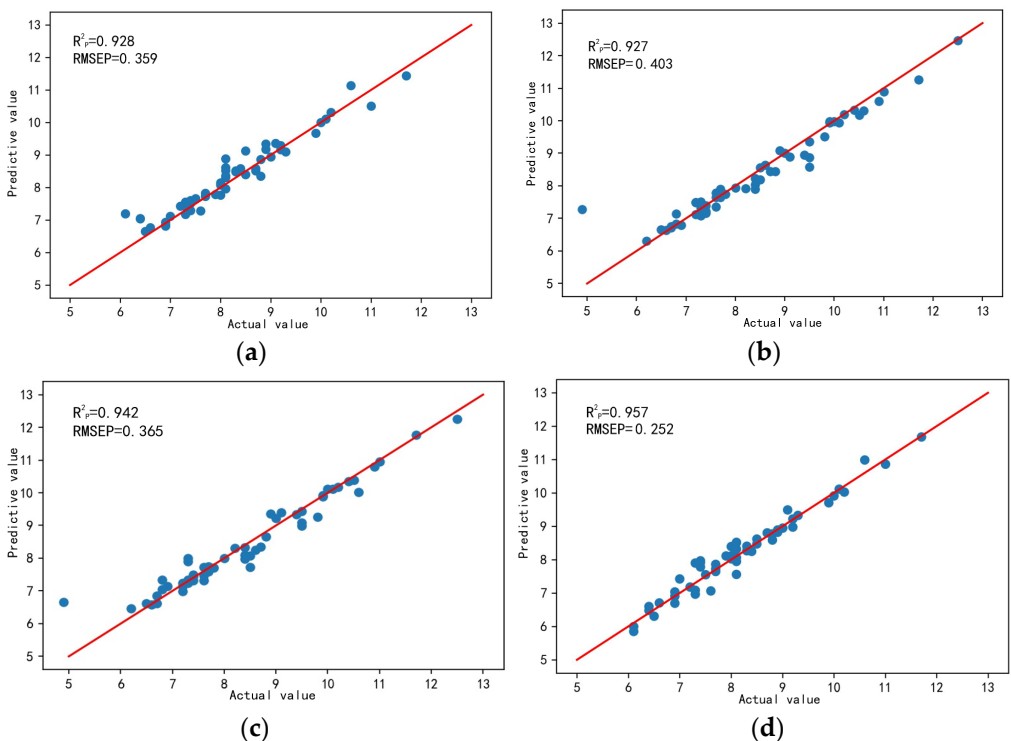

**Figure 8.** SSC prediction results of different regression models. (**a**) PLSR. (**b**) XGBoost. (**c**) CatBoost. (**d**) BO-CatBoost.

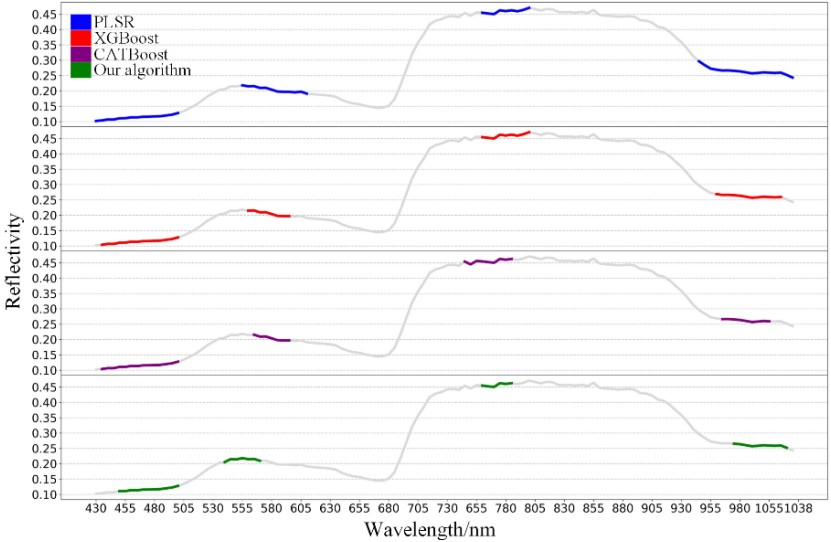

**Figure 9.** Different model selection for predicting SSC feature bands.

## 5. Conclusions

SSC prediction of green plums was taken as the research object in this paper, and the VIS-NIR and SW-NIR full-band spectra information of green plums was collected. The spectral data were calibrated and pre-processed. An improved IRS-RF algorithm was proposed to screen four groups of characteristic bands. A BO-CatBoost model based on Bayesian optimization algorithm was constructed to study SSC prediction of green plums. The main conclusions include:

(1) MSC + IRS-RF was used to preprocess the VIS-NIR spectral band and select the characteristic wavelength. The BO-CatBoost model based on Bayesian optimization algorithm outperformed PLSR, XGBoost, and CatBoost regression models in SSC prediction, with $R^2_P$ of 0.957, which was 3.1% higher than the traditional PLSR.

(2) Based on the MSC + IRS-RF + BO-CatBoost model proposed in this article, when predicting SSC values, the four selected feature band dimensions were the lowest, only 31, all less than PLSR, XGBoost, and CatBoost's 53, 43, and 39. The selected band ranges were 452–498, 538–566, 756–782, and 982–1032.

Sandra [22] used PLSR to establish a prediction model for nectarine maturity index (RPI) and internal quality index (IQI), and the results showed that the determination coefficient $R^2$ was greater than 0.87. Yang [23] used the CARS-PLSR model to predict the SSC of multi-variety tomatoes. Performances were Tianci-595, Rp was 0.85, Xianke-No. 8 Rp was 0.87, and Yuanwei-No. 1 Rp was 0.87. Zhang [24] used partial least squares (PLS) and least square-support vector machines (LS-SVM) to build the prediction models to evaluate SSC in tomatoes. The prediction results revealed that the best performance was obtained using the PLS model with the optimal wavelengths selected by CARS in the range of 900–1400 nm, and the Rp was 0.820. It could be seen that the prediction results of the optimization model proposed in this article were significantly better than the above research.

Through the above research, the model prediction accuracy met the actual sorting requirements, and the selected feature wavelength groups provide a theoretical foundation for later research on green plum sorting based on multispectral technology. Subsequently, based on the selected feature wavelength groups, a multispectral acquisition system will be established and further optimized to reduce the cost of green plum sorting, ensure sorting accuracy, and fill the research gap.

**Author Contributions:** Conceptualization, X.Z. and Y.L.; methodology, X.Z., C.Z. and Z.Z.; software, C.Z. and Q.S.; validation, Q.S. and Z.Z.; resources, X.Z., Y.Y. and Z.Z.; writing—original draft preparation, X.Z.; writing—review and editing, Y.L., C.Z. and Y.Y.; funding acquisition, Y.L. All authors have read and agreed to the published version of the manuscript.

**Funding:** This research was funded by the Jiangsu Agricultural Science and Technology Innovation Fund Project (Funding number: CX (18)3071, Funder: Jiangsu provincial department of science and technology). LIU YING. Research on key technologies of intelligent sorting for green plum.

**Institutional Review Board Statement:** "Not applicable" for studies not involving humans or animals.

**Data Availability Statement:** The experiment is not yet completed, and so, the data are not public.

**Acknowledgments:** The authors would like to extend their sincere gratitude for the technical support from the Jiangsu Co-Innovation Center of Efficient Processing and Utilization of Forest Resources.

**Conflicts of Interest:** The authors declare no conflict of interest.

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
