# Peer review of "Prediction of Solid Soluble Content of Green Plum Based on Improved CatBoost"

_agriculture, doi:10.3390/agriculture13061122_

Round 1
Reviewer 1 Report
The manuscript is based on a simple but thorough analysis of spectral information and can be further used in fruit post harvest processing. There are some parts that need to be improved.
In the introduction section, many individual comments are inserted but I will here highlight those the most important. Please define "green plums". These fruits belong to which botanical species, cultivar etc. The first two paragraphs of Introduction section are quite hard to understand, especially because of English.
Also, Materials and methods section needs more information about samples of plums. Was there 276 fruits, were there any repetitions, what are training and test sets?

Some sentences are quite hard to understand. Please revise English thru the text, especially the Introduction.
Author Response
Thank you very much for your comments and suggestions,and my response as follows,
1.In the introduction section, many individual comments are inserted but I will here highlight those the most important.
Response: The most important comments have been highlighted in the manuscript.
2.Please define "green plums". These fruits belong to which botanical species, cultivar etc.
Response: The specie of green plums is Zhao Shui green plum, and the green plum samples were purchased and screened from Yunnan Province, China.
3.The first two paragraphs of Introduction section are quite hard to understand, especially because of English.
Response: The first two paragraphs of introduction section have been revised.
4.Also, Materials and methods section needs more information about samples of plums. Was there 276 fruits, were there any repetitions, what are training and test sets?
Response: The information about samples of plums has been added in the manuscript. The number of green plum samples was 276 for the purpose of SSC prediction. The samples were divided into a training set and a test set in a ratio of about 4:1, with 221 samples as the training set and 55 samples as the test set. The training set is used to establish the model, and the testing set is used to verify the performance of the final selection of the optimal model.
Reviewer 2 Report
Manuscript presents an interesting approach for phenotyping od plum fruits aimed to their sorting and harvest. The authors use imagining technique allowing to collect spectra which are further analysed using a few multivariate methods. Results, are in fact, a comparison of several sophisticated methods to show their relative superiority, on the base of exploration of VIS-NIR and SW-NIR reflectance spectra
The authors show evident high competence in modern techniques applicable for phenotyping of biological material and statistical analysis using advanced supervised models.
I have reservation whether this manuscript is appropriate for the "agriculture" because of its highly sophisticated tools used and focussing only on selection of appropriate model.
The abstract is too much technical and cannot be understood by a reader who is not deeply rooted in the issue and methodology.
I would suggest submission to other journal more focused on statistical modelling and directed to readers interested in statistical modelling. It may be more suited e.g to “sensors” journal.
Author Response
Thank you very much for your comments and suggestions,and my response as follows,
1.The abstract is too much technical and cannot be understood by a reader who is not deeply rooted in the issue and methodology.
Response: The abstract has been revised to be easily understood by a reader who is not deeply rooted in the issue and methodology.
Reviewer 3 Report
Ref: MS entitled “Research on Results Prediction of SSC of Green Plum Based on Improved Catboost of Multi-channel Band Groups”. The work is very interesting and contains useful information with practical applicability in assessing the soluble solids content of green plum. The reported information could be used practically for the maturity assessment in terms of SSC content. I have some comments and suggestions for the improvement of the article. It is requested to carefully check the abbreviated terms as some have mentioned without their full form at some places of the manuscript. Please check in the whole article. Please supplement some information regarding SSC prediction and usefulness in assessing maturity in the introduction. In addition, please improve the work objectives. Please improve quality of the figure 1 by increasing font size. What was the cultivar name? Also, how many cultivars tested either only one or more than one? The background of the figure 3 needs to be improved so that the cultivar becomes very clear in sight. The results are well written but I could not see any suitable references in the discussion part. Please see carefully and supplement related literature in the discussion and compare it with other related fruits. The technical names in the references are not in italic form.
Only minor editing is needed.
Author Response
Thank you very much for your comments and suggestions,and my response as follows,
1.It is requested to carefully check the abbreviated terms as some have mentioned without their full form at some places of the manuscript. Please check in the whole article.
Response: The abbreviated terms has been carefully checked and revised in the whole article.
2.Please supplement some information regarding SSC prediction and usefulness in assessing maturity in the introduction.
Response: The information regarding SSC prediction and usefulness in assessing maturity has been supplemented in the introduction.
3.In addition, please improve the work objectives. Please improve quality of the figure 1 by increasing font size.
Response:
The font size of figure 1 has been increased.
The work objective is to resolve the problems of time-consuming, high cost, and difficulty in non-destructive testing technology caused by high-spectrum equipment, and make model prediction accuracy meet the actual sorting requirements. The selected feature band groups provide a theoretical basis for future multi-spectral technology based on green plum sorting research. It has been improved.
4.What was the cultivar name? Also, how many cultivars tested either only one or more than one?
Response: The specie of green plums is Zhao Shui green plum, and the green plum samples were purchased and screened from Yunnan Province, China.
The number of green plum samples was 276 for the purpose of SSC prediction. The samples were divided into a training set and a test set in a ratio of about 4:1, with 221 samples as the training set and 55 samples as the test set.
5.The background of the figure 3 needs to be improved so that the cultivar becomes very clear in sight.
Response: Figure 3 are the original pseudo color images generated by the camera based on hyperspectral data.
6.The results are well written but I could not see any suitable references in the discussion part. Please see carefully and supplement related literature in the discussion and compare it with other related fruits.
Response: Related literature has been supplemented and compared with other related fruits in the discussion part.
7.The technical names in the references are not in italic form.
Response: The technical names in the references has been revised.
Reviewer 4 Report
Dear authors
thank you for giving me the opportunity to review the manuscript. It is interesting but have many drawbacks to be improved
Abbreviations are used in the title and abstract without mentioning at first
The title is confusing and must be changed to explain why the research was done
The main objectives of the research, and the hypothesis must be clear
Probably the title must be "Prediction of solid soluble content based on IRS-RF algorithm or VIS-NIR and SW-NIR images", my suggestion
The manuscript is confusing, are the authors testing different models? or comparing?
The authors say that fruits will be collected, but the experiment was just done. Please use the proper tense
Material and Methods are not clear
how many measurements were done for each fruit?
Which physical and chemical experiments do the authors refer to?
Lines 115 to 119, change de present tense. The same to 126, 131- 136, 139 to 142
item 2.3: It is not an M & M. Authors must state how RF feature extraction was performed for this research and the applicability of the method
Again on item 3: Research on regression.... why this subtitle?
In summary, the materials and methods are poorly described.
Conclusions are not conclusions (268-274).
Improve the use of past tense. In many sections authors use present or future tense.
Author Response
Thank you very much for your comments and suggestions,and my response as follows,
1.Abbreviations are used in the title and abstract without mentioning at first
Response: The abbreviated terms has been carefully checked and revised in the whole article.
2.The title is confusing and must be changed to explain why the research was done
Response: The title has been changed as “Prediction of Solid Soluble Content of Green Plum Based on Improved Catboost”.
3.The main objectives of the research, and the hypothesis must be clear
Response: The work objective is to resolve the problems of time-consuming, high cost, and difficulty in non-destructive testing technology caused by high-spectrum equipment, and make model prediction accuracy meet the actual sorting requirements. The selected feature band groups provide a theoretical basis for future multi-spectral technology based on green plum sorting research. It has been improved.
4.Probably the title must be "Prediction of solid soluble content based on IRS-RF algorithm or VIS-NIR and SW-NIR images", my suggestion
Response: The title has been changed as “Prediction of Solid Soluble Content of Green Plum Based on Improved Catboost”.
5.The manuscript is confusing, are the authors testing different models? or comparing?
Response: Random forest algorithm based on induced random selection(IRS-RF) is proposed to screen 4 sets of characteristic wavebands. Bayesian optimization CatBoost model (BO-CatBoost) is constructed to predict SSC value of green plums.
The experimental results show that the model proposed in this paper based on MSC+IRS-RF+BO-CatBoost is superior to traditional PLSR, XGBoost, and CatBoost in predicting SSC, with R2P of 0.957, which is 3.1% higher than PLSR.
6.The authors say that fruits will be collected, but the experiment was just done. Please use the proper tense
Response: The proper tense has been revised in the article.
7.Material and Methods are not clear
Response: The materials and methods has been redescribed.
how many measurements were done for each fruit?
Response: Each green plum sample was measured only once. After spectral data was collected , the green plums were squeezed to extract green plum juice immediately. The PAL-1 hand-held refractometer was used to measure the SSC of green plums. The measurement range was 0.0-53.0% BRIX, with an accuracy of ±0.2% BRIX. The sample tank should be cleaned before measuring, then the green plum sample was squeezed into juice.After precipitating, the supernatant was dropped into the sample tank, the SSC value was recorded.
Which physical and chemical experiments do the authors refer to?
Response: It has been changed as “SSC value determination of each green plum”.
8.Lines 115 to 119, change de present tense. The same to 126, 131- 136, 139 to 142
Response: The proper tense has been revised.
9.item 2.3: It is not an M & M. Authors must state how RF feature extraction was performed for this research and the applicability of the method
Response: The RF feature extraction has been explained in item 2.3.
10.Again on item 3: Research on regression.... why this subtitle?
Response: The subtitle has been deleted.
11.In summary, the materials and methods are poorly described.
Response: The materials and methods has been redescribed.
12.Conclusions are not conclusions (268-274).
Response: The conclusions part has been redescribed.
13.Improve the use of past tense. In many sections authors use present or future tense.
Response: The use of past tense has been improved in the whole article.
Round 2
Reviewer 2 Report
I found substantial improvement of the manuscript, so this version can be recommended for publication